# EphB2-Targeting Monoclonal Antibodies Exerted Antitumor Activities in Triple-Negative Breast Cancer and Lung Mesothelioma Xenograft Models

**DOI:** 10.3390/ijms26178302

**Published:** 2025-08-27

**Authors:** Rena Ubukata, Tomokazu Ohishi, Mika K. Kaneko, Hiroyuki Suzuki, Yukinari Kato

**Affiliations:** 1Department of Antibody Drug Development, Tohoku University Graduate School of Medicine, 2-1 Seiryo-machi, Aoba-ku, Sendai 980-8575, Miyagi, Japan; ubukata.rena.p7@dc.tohoku.ac.jp (R.U.); mika.kaneko.d4@tohoku.ac.jp (M.K.K.); 2Institute of Microbial Chemistry (BIKAKEN), Laboratory of Oncology, Microbial Chemistry Research Foundation, 3-14-23 Kamiosaki, Shinagawa-ku, Tokyo 141-0021, Japan; ohishit@bikaken.or.jp

**Keywords:** EphB2, monoclonal antibody, antibody-dependent cellular cytotoxicity, complement-dependent cytotoxicity, triple-negative breast cancer

## Abstract

Eph receptor B2 (EphB2) overexpression is associated with poor clinical outcomes in various tumors. EphB2 is involved in malignant tumor progression through the promotion of invasiveness and metastasis. Genetic and transcriptome analyses implicated that EphB2 is a therapeutic target for specific tumor types. A monoclonal antibody (mAb) is one of the essential therapeutic strategies for EphB2-positive tumors. We previously developed an anti-EphB2 mAb, Eb_2_Mab-12 (IgG_1_, kappa), by immunizing mice with EphB2-overexpressed glioblastoma. Eb_2_Mab-12 specifically reacted with the EphB2-overexpressed Chinese hamster ovary-K1 (CHO/EphB2) and some cancer cell lines in flow cytometry. In this study, we engineered Eb_2_Mab-12 into a mouse IgG_2a_ type (Eb_2_Mab-12-mG_2a_) and a human IgG_1_-type (Eb_2_Mab-12-hG_1_) mAb. Eb_2_Mab-12-mG_2a_ and Eb_2_Mab-12-hG_1_ retained the reactivity to EphB2-positive cells and exerted antibody-dependent cellular cytotoxicity and complement-dependent cytotoxicity in the presence of effector cells and complements, respectively. In CHO/EphB2, triple-negative breast cancer, and lung mesothelioma xenograft models, both Eb_2_Mab-12-mG_2a_ and Eb_2_Mab-12-hG_1_ exhibited potent antitumor efficacy. These results indicated that Eb_2_Mab-12-derived mAbs could be applied to mAb-based therapy against EphB2-positive tumors.

## 1. Introduction

The mammalian ephrin–Eph system comprises eight membrane-bound ephrin ligands, including five glycosylphosphatidylinositol (GPI)-anchored ephrin-As and three transmembrane ephrin-Bs, and 14 receptor tyrosine kinases (RTKs), including nine EphA and five EphB receptors [1,2,3,4,5,6]. Upon ligand binding, Eph receptors undergo dimerization or oligomerization, resulting in the autophosphorylation of specific tyrosine residues within the cytoplasmic domains of both the Eph receptors and ephrin-B ligands [7]. These phosphorylated tyrosine residues serve as docking sites for downstream cytoplasmic signaling molecules containing Src homology 2 (SH2), PDZ, or phosphotyrosine-binding (PTB) domains [8]. Consequently, Eph–ephrin interactions initiate bidirectional signaling: forward signaling through Eph receptors and reverse signaling through ephrin-Bs, which is critical for intercellular communication between identical or distinct cell types [2,6,9]. The bidirectional signaling by the Eph–ephrin axis regulates various biological processes, including tissue morphogenesis, regeneration, and homeostasis. Dysregulation of the system has been implicated in multiple diseases such as cancer [3,4].

The Eph system exerts context-dependent functions in tumorigenesis, acting either as a tumor promoter or suppressor depending on the cellular context [10]. Among Eph receptors, EphB2 is overexpressed in various tumors, including breast cancer [11], glioblastoma [12], mesothelioma [13], and hepatocellular carcinoma [14], which correlates with poor clinical prognosis. EphB2 has been shown to promote tumor cell migration and invasion through forward signaling [15,16]. Therefore, EphB2 is believed to act as an oncogene in those tumors. Given their roles in tumor progression, EphB2 has emerged as a promising target for monoclonal antibody (mAb)-based therapeutic strategies [17,18,19,20,21,22,23,24].

An anti-EphB2 mAb (clone 2H9) effectively blocked the interaction of EphB2 with ephrin-B2 and inhibited the EphB2 autophosphorylation [25]. However, 2H9 did not affect the proliferation of EphB2-positive tumors [25]. Therefore, 2H9 was further developed into an antibody-drug conjugate (ADC) to monomethylauristatin E and showed the antitumor efficacy to EphB2-overexpressed fibrosarcoma HT1080 and colorectal cancer xenografts [25]. However, the 2H9-ADC was not further developed in the clinic [26].

Triple-negative breast cancer (TNBC) is defined by the lack of expression of estrogen receptor (ER), progesterone receptor (PR), and human epidermal growth factor receptor 2 (HER2)—key molecular markers that are routinely utilized for breast cancer classification and therapeutic decision-making [27]. Compared to other breast cancer subtypes, TNBC is characterized by a more aggressive clinical course, including higher rates of recurrence and distant metastasis, resulting in a poor overall survival [28]. Due to the intertumoral heterogeneity, TNBC is further subclassified into distinct molecular subtypes, which facilitates the development of more targeted therapeutic strategies. Bioinformatics and transcriptomic profiling have enabled the stratification of TNBC into four intrinsic molecular subtypes: basal-like 1 (BL1), basal-like 2 (BL2), mesenchymal (M), and luminal androgen receptor (LAR). Each of these subtypes exhibits unique molecular signatures, biological behaviors, and clinical outcomes [29,30]. Among the intrinsic TNBC subtypes, the BL2 subtype is associated with the highest risk of disease progression and the poorest clinical outcomes. Patients with BL2 tumors exhibit the shortest median overall survival [31], and demonstrate minimal responsiveness to neoadjuvant chemotherapy. EphB2 was identified as one of the signature tyrosine kinases of the BL2 subtype, which could lead to novel approaches for tumor diagnosis and targeted therapy [29,32].

Ependymoma is a common type of brain tumor in children and arises in the supratentorial region, spinal cord, or posterior fossa [33]. Extensive molecular analyses of ependymoma have clarified distinct molecular profiles despite similar histology. A pioneering study analyzed the alterations of DNA copy number and revealed the EphB2 amplification and INK4A/ARF deletion in supratentorial ependymoma [34]. Recently, supratentorial ependymoma has been further categorized into EPN-ZFTA, EPN-YAP1, and subependymoma based on their genetic profile [35]. A fusion protein EPN-ZFTA binds to the proximal enhancers of EphB2, which leads to aberrant expression of EphB2 and tumor progression [36]. Therefore, EphB2 is a context-dependent oncoprotein and a potential therapeutic target for treating those tumors.

For targeting Eph receptors, we have developed mAbs against Eph receptors, including EphA2 (clone Ea_2_Mab-7) [37], EphA3 (clone Ea_3_Mab-20) [38], EphB2 (clone Eb_2_Mab-12) [39], EphB4 (clone B_4_Mab-7) [40], and EphB6 (clone Eb_6_Mab-3) [41] using the Cell-Based Immunization and Screening (CBIS) method. Among them, Eb_2_Mab-12 (mouse IgG_1_, kappa) recognized EphB2-positive cells with high affinity. Furthermore, Eb_2_Mab-12 did not show the cross-reactivity to other EphA and EphB receptors [39]. Therefore, Eb_2_Mab-12 has potential to the application for tumor therapy.

In this study, we engineered Eb_2_Mab-12 into a mouse IgG_2a_-type (Eb_2_Mab-12-mG_2a_) and a human IgG_1_-type (Eb_2_Mab-12-hG_1_) mAbs, and evaluated antibody-dependent cellular cytotoxicity (ADCC), complement-dependent cytotoxicity (CDC), and antitumor efficacy in EphB2-positive tumor xenograft models.

## 2. Results

### 2.1. Production of Mouse IgG_2a_-Type and Human IgG_1_-Type Anti-EphB2 mAbs from Eb_2_Mab-12

We previously generated a mAb against EphB2, designated Eb_2_Mab-12 (mouse IgG_1_, κ), by immunizing mice with EphB2-overexpressed glioblastoma LN229. Eb_2_Mab-12 showed a high binding affinity and specificity among 14 Eph receptors [39]. In this study, we first determined the complementarity-determining regions (CDRs) of Eb_2_Mab-12 and produced a recombinant Eb_2_Mab-12 (mouse IgG_1_) (Figure 1A). To evaluate the antitumor activity, a mouse IgG_2a_-type Eb_2_Mab-12 (Eb_2_Mab-12-mG_2a_) was generated by fusing the V_H_ and V_L_ CDRs of Eb_2_Mab-12 with the C_H_ and C_L_ chains of mouse IgG_2a_ (Figure 1A). Furthermore, a human IgG_1_-type Eb_2_Mab-12 (Eb_2_Mab-12-hG_1_) was generated by fusing the V_H_ and V_L_ CDRs of Eb_2_Mab-12 with the C_H_ and C_L_ chains of human IgG_1_ (Figure 1A). As a control mouse IgG_2a_ (mIgG_2a_) and a human IgG_1_ (hIgG_1_) mAb, we previously produced PMab-231 (an anti-tiger podoplanin mAb, mouse IgG_2a_) [42] and humCvMab-62 (an anti-SARS-CoV-2 spike protein S2 subunit mAb, human IgG_1_) [43], respectively. We confirmed the purity of the recombinant mAbs by SDS-PAGE under reduced conditions (Figure 1B).

### 2.2. Flow Cytometry Using Eb_2_Mab-12-mG_2a_ and Eb_2_Mab-12-hG_1_

We next confirmed the reactivity of Eb_2_Mab-12-mG_2a_ and Eb_2_Mab-12-hG_1_ to CHO/EphB2 and endogenous EphB2-positive cancer cell lines. As shown in Figure 2A, Eb_2_Mab-12-mG_2a_ and Eb_2_Mab-12-hG_1_ showed the comparable reactivity to CHO/EphB2 with Eb_2_Mab-12. Control mIgG_2a_ and hIgG_1_ did not recognize CHO/EphB2 (Figure 2A). In contrast, Eb_2_Mab-12-mG_2a_, Eb_2_Mab-12-hG_1_, and Eb_2_Mab-12 did not react with parental CHO-K1 (Figure 2B). We previously tested the expression of EphB2 in more than 100 cell lines using flow cytometry and found that breast cancer MDA-MB-231, lung mesothelioma NCI-H226, and colorectal cancer LS174T showed the EphB2 expression [39]. Eb_2_Mab-12-mG_2a_ and Eb_2_Mab-12-hG_1_ exhibited the comparable reactivity to MDA-MB-231 (Figure 2C) and NCI-H226 (Figure 2D) with Eb_2_Mab-12. Control mIgG_2a_ and hIgG_1_ did not recognize MDA-MB-231 (Figure 2C) and NCI-H226 (Figure 2D). We also confirmed the expression of EphB2 in LS174T (Appendix A). We next used EphB2-knockout LS174T (BINDS-58). Eb_2_Mab-12-mG_2a_ and Eb_2_Mab-12-hG_1_ reacted with LS174T compared to each isotype control, but did not react with BINDS-58 (Appendix A). These results indicated that Eb_2_Mab-12-mG_2a_ and Eb_2_Mab-12-hG_1_ retain the reactivity to EphB2-positive cells.

### 2.3. ADCC and CDC by Eb_2_Mab-12-mG_2a_ and Eb_2_Mab-12-hG_1_ Against CHO/EphB2

We next examined the ADCC caused by Eb_2_Mab-12-mG_2a_ and Eb_2_Mab-12-hG_1_ against CHO/EphB2. We used the splenocytes derived from BALB/c nude mice as an effector since human IgG_1_ can bind to all four mouse FcγRs and exert ADCC in the presence of mouse effector cells [44]. Eb_2_Mab-12-mG_2a_ showed ADCC against CHO/EphB2 (30.4% vs. 8.3% cytotoxicity of control mIgG_2a_, *p* < 0.05, Figure 3A left). Eb_2_Mab-12-hG_1_ also exerted ADCC against CHO/EphB2 (45.5% vs. 16.1% cytotoxicity of control hIgG_1_, *p* < 0.05, Figure 3A right). Next, we examined CDC caused by Eb_2_Mab-12-mG_2a_ and Eb_2_Mab-12-hG_1_ against CHO/EphB2 in the presence of complements. Eb_2_Mab-12-mG_2a_ elicited CDC against CHO/EphB2 (45.1% vs. 15.8% cytotoxicity of control mIgG_2a_, *p* < 0.05, Figure 3B left). Eb_2_Mab-12-hG_1_ elicited similar CDC against CHO/EphB2 (40.6% vs. 17.9% cytotoxicity of control hIgG_1_, *p* < 0.05, Figure 3B right). These results showed that Eb_2_Mab-12-mG_2a_ and Eb_2_Mab-12-hG_1_ exhibited potent ADCC and CDC against CHO/EphB2.

### 2.4. Antitumor Effect by Eb_2_Mab-12-mG_2a_ and Eb_2_Mab-12-hG_1_ Against CHO/EphB2 Xenografts

In preclinical studies of mAbs such as trastuzumab (human IgG_1_), the antitumor effect was proved using human breast cancer xenografts in nude mice without human-derived effectors [45,46,47]. We then examined the antitumor effect of Eb_2_Mab-12-mG_2a_ and Eb_2_Mab-12-hG_1_ against CHO/EphB2 xenografts inoculated in nude mice. Following the inoculation of CHO/EphB2, Eb_2_Mab-12-mG_2a_ or control mIgG_2a_ was intraperitoneally injected into CHO/EphB2 xenograft-bearing mice on days 8 and 14. Eb_2_Mab-12-hG_1_ or control hIgG_1_ was also injected as described above. The tumor volume was measured on days 8, 10, 14, 16, and 21 after the inoculation. The Eb_2_Mab-12-mG_2a_ administration resulted in a potent and significant reduction in CHO/EphB2 xenografts on days 16 (*p* < 0.01) and 21 (*p* < 0.01) compared with that of mIgG_2a_ (Figure 4A). The Eb_2_Mab-12-hG_1_ administration also showed a significant reduction in CHO/EphB2 xenografts on day 21 (*p* < 0.01) compared with that of hIgG_1_ (Figure 4B). Significant decreases in xenograft weight caused by Eb_2_Mab-12-mG_2a_ and Eb_2_Mab-12-hG_1_ were observed in CHO/EphB2 xenografts (87% reduction; *p* < 0.01; Figure 4C and 77% reduction; *p* < 0.01; Figure 4D, respectively). Body weight loss was not observed in the xenograft-bearing mice treated with Eb_2_Mab-12-mG_2a_ (Figure 4E) and Eb_2_Mab-12-hG_1_ (Figure 4F).

### 2.5. ADCC and CDC by Eb_2_Mab-12-mG_2a_ and Eb_2_Mab-12-hG_1_ Against Endogenous EphB2-Positive Cancer Cell Lines

We next investigated ADCC caused by Eb_2_Mab-12-mG_2a_ and Eb_2_Mab-12-hG_1_ against the endogenous EphB2-positive cancer cell lines in the presence of splenocytes derived from BALB/c nude mice. Eb_2_Mab-12-mG_2a_ showed ADCC against MDA-MB-231 (9.2% vs. 3.2% cytotoxicity of control mIgG_2a_, *p* < 0.05, Figure 5A left). Eb_2_Mab-12-hG_1_ also exerted ADCC against MDA-MB-231 (11.0% vs. 3.0% cytotoxicity of control hIgG_1_, *p* < 0.05, Figure 5A right). We then examined CDC caused by Eb_2_Mab-12-mG_2a_ and Eb_2_Mab-12-hG_1_ against MDA-MB-231 in the presence of complements. Eb_2_Mab-12-mG_2a_ elicited CDC against MDA-MB-231 (3.2% vs. 0.80% cytotoxicity of control mIgG_2a_, *p* < 0.05, Figure 5B left). Eb_2_Mab-12-hG_1_ also elicited CDC against MDA-MB-231 (3.8% vs. 1.3% cytotoxicity of control hIgG_1_, *p* < 0.05, Figure 5B right).

Eb_2_Mab-12-mG_2a_ showed ADCC against NCI-H226 (16.8% vs. 6.5% cytotoxicity of control mIgG_2a_, *p* < 0.05, Figure 5C left). Eb_2_Mab-12-hG_1_ also exerted ADCC against NCI-H226 (12.5% vs. 4.6% cytotoxicity of control hIgG_1_, *p* < 0.05, Figure 5C right). We then examined CDC caused by Eb_2_Mab-12-mG_2a_ and Eb_2_Mab-12-hG_1_ against NCI-H226 in the presence of complements. Eb_2_Mab-12-mG_2a_ elicited CDC against NCI-H226 (12.1% vs. 5.4% cytotoxicity of control mIgG_2a_, *p* < 0.05, Figure 5D left). Eb_2_Mab-12-hG_1_ also elicited CDC against NCI-H226 (7.4% vs. 1.6% cytotoxicity of control hIgG_1_, *p* < 0.05, Figure 5D right).

We also assessed ADCC and CDC caused by Eb_2_Mab-12-mG_2a_ and Eb_2_Mab-12-hG_1_ against LS174T. However, we could not observe the significant ADCC by both mAbs and CDC by Eb_2_Mab-12-mG_2a_. Only a significant CDC by Eb_2_Mab-12-hG_1_ was observed (Appendix A). These results showed that Eb_2_Mab-12-mG_2a_ and Eb_2_Mab-12-hG_1_ exhibited significant ADCC and CDC against MDA-MB-231 and NCI-H226.

### 2.6. Antitumor Effect by Eb_2_Mab-12-mG_2a_ and Eb_2_Mab-12-hG_1_ Against Endogenous EphB2-Positive Cancer Xenografts

The antitumor activity of Eb_2_Mab-12-mG_2a_ and Eb_2_Mab-12-hG_1_ against MDA-MB-231 xenograft was investigated. Following the inoculation of the MDA-MB-231, Eb_2_Mab-12-mG_2a_ or control mIgG_2a_ was intraperitoneally injected into MDA-MB-231 xenograft-bearing mice on days 8 and 14. Eb_2_Mab-12-hG_1_ or control hIgG_1_ was also injected as described above. The tumor volume was measured on days 8, 10, 14, 16, 21, and 24 after the inoculation. The Eb_2_Mab-12-mG_2a_ administration resulted in a significant reduction in MDA-MB-231 xenografts on days 21 (*p* < 0.01) and 24 (*p* < 0.01) compared with that of mIgG_2a_ (Figure 6A). The Eb_2_Mab-12-hG_1_ administration also showed a significant reduction in MDA-MB-231 xenografts on days 21 (*p* < 0.01) and 24 (*p* < 0.01) compared with that of hIgG_1_ (Figure 6B). Significant reductions in xenograft weight caused by Eb_2_Mab-12-mG_2a_ and Eb_2_Mab-12-hG_1_ were observed in MDA-MB-231 xenografts (33% reduction; *p* < 0.05; Figure 6C and 23% reduction; *p* < 0.05; Figure 6D, respectively). Body weight loss was not observed in the xenograft-bearing mice treated with Eb_2_Mab-12-mG_2a_ (Figure 6E) and Eb_2_Mab-12-hG_1_ (Figure 6F).

The antitumor activity of Eb_2_Mab-12-mG_2a_ and Eb_2_Mab-12-hG_1_ against NCI-H226 xenograft was investigated. Following the inoculation of the NCI-H226, Eb_2_Mab-12-mG_2a_ or control mIgG_2a_ was intraperitoneally injected into NCI-H226 xenograft-bearing mice on days 7 and 14. Eb_2_Mab-12-hG_1_ or control hIgG_1_ was also injected as described above. The tumor volume was measured on days 7, 10, 14, 17, and 21 after the inoculation. The Eb_2_Mab-12-mG_2a_ administration resulted in a significant reduction in NCI-H226 xenografts on days 21 (*p* < 0.01) compared with that of mIgG_2a_ (Figure 7A). The Eb_2_Mab-12-hG_1_ administration also showed a significant reduction in NCI-H226 xenografts on days 21 (*p* < 0.01) compared with that of hIgG_1_ (Figure 7B). Significant reductions in xenograft weight caused by Eb_2_Mab-12-mG_2a_ and Eb_2_Mab-12-hG_1_ were observed in NCI-H226 xenografts (29% reduction; *p* < 0.01; Figure 7C and 33% reduction; *p* < 0.01; Figure 7D, respectively). Body weight loss was not observed in the xenograft-bearing mice treated with Eb_2_Mab-12-mG_2a_ (Figure 7E) and Eb_2_Mab-12-hG_1_ (Figure 7F).

In the same way of ADCC and CDC, the obvious antitumor activity of Eb_2_Mab-12-mG_2a_ and Eb_2_Mab-12-hG_1_ against LS174T xenograft was not observed (Appendix A).

## 3. Discussion

A growing body of evidence has suggested that EphB2 mediates malignant progression in various types of tumors and is a candidate for therapeutic targets and diagnostic biomarkers. Genetic and transcriptome analyses revealed that EphB2 is amplified [34] or upregulated by EPN-ZFTA in supratentorial ependymoma [36], and EphB2 is a signature tyrosine kinase of the BL2 subtype in TNBC [29,32]. In this study, we demonstrated that Eb_2_Mab-12-mG_2a_ and Eb_2_Mab-12-hG_1_ retain the reactivity to EphB2-positive cells (Figure 2) and exert ADCC and CDC in the presence of mouse splenocytes and complements, respectively (Figure 3 and Figure 5). Eb_2_Mab-12-mG_2a_ and Eb_2_Mab-12-hG_1_ showed the antitumor effect against xenograft tumors of CHO/EphB2 (Figure 4), TNBC breast cancer MDA-MB-231 (Figure 6), and lung mesothelioma NCI-H226 (Figure 7). These results indicated that Eb_2_Mab-12-derived mAbs could be applied to antibody-based therapy against EphB2-positive tumors.

The BL2 subtype of TNBC is characterized by the enrichment in growth factor signaling and myoepithelial markers [27]. EphB2 is identified as a potential differentiator of BL2 with EphA4, receptor tyrosine kinase-like orphan receptor 1, platelet-derived growth factor receptor (PDGFR)A, PDGFRB, and epidermal growth factor receptor (EGFR) [29,32]. MDA-MB-231 cells exhibited the highest sensitivity to the multikinase inhibitor dasatinib [27], which inhibits PDGFRA, PDGFRB, and EphB2 with IC_50_ values below 1 nM [48]. These results suggest that MDA-MB-231 growth depends on these signaling. Therefore, the targeting of EphB2 is an essential strategy for EphB2-positive TNBC therapy. Since Eb_2_Mab-12 is not suitable for immunohistochemistry [39], a reliable mAb should be developed for the detection of EphB2 in immunohistochemistry. For TNBC treatment, anti-trophoblast cell-surface antigen 2 (TROP2) ADCs (sacituzumab govitecan-hziy and datopotamab deruxtecan) are currently approved by the U.S. Food and Drug Administration [49]. TROP2 expression is highest in TNBC compared to other types of breast cancer. The high TROP2 expression was associated with higher androgen receptor expression, ductal carcinoma in situ, apocrine histology, lymphovascular invasion, and lymph node involvement [50]. Further studies are essential to clarify the relationship between TROP2 and EphB2 expression in TNBC subtypes. Additionally, we have developed ADCs using previously established mAbs such as anti-podoplanin mAbs [51]. We would like to develop Eb_2_Mab-12-ADCs and evaluate the antitumor activities in the future.

As shown in Appendix A, Eb_2_Mab-12-mG_2a_ and Eb_2_Mab-12-hG_1_ showed superior reactivity to LS174T compared to that of MDA-MB-231 and NCI-H226 in flow cytometry. However, Eb_2_Mab-12-mG_2a_ and Eb_2_Mab-12-hG_1_ did not exhibit the ADCC (Appendix A) and antitumor efficacy against the LS174T xenograft. The mice harboring LS174T xenograft were euthanized on day 17 after 2 times treatments of mAbs because tumors approached the limits of acceptable tumor burden (Appendix A). Previously, antitumor effects against LS174T xenograft were evaluated using T cell-engaging anti-EGFR bispecific antibody (anti-EGFR BiTE) compared to anti-EGFR mAb, cetuximab [52]. Although the anti-EGFR BiTE showed a potent antitumor effect against pancreatic adenocarcinoma COLO 356/FG, it was not effective for LS174T xenograft growth compared to cetuximab [52]. Since cetuximab possesses the neutralized ability to EGFR [53], the neutralization effect may be superior to rapid-growing LS174T xenograft compared to antitumor immunity by anti-EGFR BiTE. In our experimental method, the antitumor immunity by Eb_2_Mab-12-mG_2a_ and Eb_2_Mab-12-hG_1_ could not keep up with the rapid growth of LS174T xenograft.

EphB2 expression was significantly high in malignant mesothelioma compared to matched normal peritoneum [13]. Silencing EphB2 in mesothelioma cell lines resulted in a pronounced reduction in downstream effectors including matrix metalloproteinase-2 and vascular endothelial growth factor, while the expression of pro-apoptotic mediators such as caspase-8 was upregulated. Importantly, EphB2 silencing was associated with an increase in apoptosis [13]. Since the neutralizing effect of Eb_2_Mab-12 has not been evaluated, further studies are needed to explore it. Additionally, we examine whether cell death was actually triggered in tumor xenograft treated with Eb_2_Mab-12-mG_2a_ or Eb_2_Mab-12-hG_1_.

In ependymomas, surgery and irradiation are still the main treatment because chemotherapy is ineffective in most patients. Consequently, ependymoma is incurable in up to 40% of cases [54]. Therefore, the development of targeted therapy is essential to cure the patients. Induction of EphB2 signaling in cerebral neural stem cells lacking the Ink4a/Arf locus led to the development of supratentorial ependymoma in mouse brain [34]. The EphB2-driven model exhibits high penetrance and recapitulates the histology and transcriptomic features of the corresponding human supratentorial ependymoma [34]. In preclinical study, a genetically engineered mouse model driven by Ephb2 has been used to evaluate the potential therapeutic targets [55]. The multikinase inhibitor dasatinib suppressed the growth of ependymoma through inhibition of EphB2 signaling. Dasatinib treatment also remodeled the ependymoma immune microenvironment by promoting the polarization of tumor-associated macrophages toward an M1-like phenotype and enhancing the activation of CD8^+^ T cells [55]. The evaluation anti-EphB2 mAbs in the preclinical model is essential to obtain the proof of concept for clinical study.

## 4. Materials and Methods

### 4.1. Cell Lines

CHO-K1, MDA-MB-231 (TNBC), NCI-H226 (lung mesothelioma), and LS174T (colorectal cancer) cell lines were obtained from the American Type Culture Collection (Manassas, VA, USA). CHO/EphB2 was established previously [39]. EphB2-knockout LS174T (BINDS-58) was generated using the CRISPR/Cas9 system with EphB2-specific guide RNA (ACTACAGCGACTGCTGAGCT). Knockout cell lines were isolated using the SH800S cell sorter based on the loss of reactivity to OptiBuild™ RB545 mouse anti-human EphB2 mAb (clone 2H9, BD Biosciences, Franklin Lakes, NJ, USA).

MDA-MB-231, NCI-H226, LS174T, and BINDS-58 were maintained in Dulbecco’s Modified Eagle’s Medium (DMEM; Nacalai Tesque, Inc., Kyoto, Japan). CHO-K1 and CHO/EphB2 were cultured in Roswell Park Memorial Institute (RPMI) 1640 medium (Nacalai Tesque, Inc.). All culture media were supplemented with 10% heat-inactivated fetal bovine serum (FBS; Thermo Fisher Scientific Inc., Waltham, MA, USA), 100 U/mL penicillin, 100 μg/mL streptomycin, and 0.25 μg/mL amphotericin B (Nacalai Tesque, Inc.). Cells were incubated at 37 °C in a humidified atmosphere containing 5% CO_2_ and 95% air.

### 4.2. Recombinant mAb Production

An anti-EphB2 mAb, Eb_2_Mab-12 (mouse IgG_1_, κ) [39] was established previously. To construct the mouse IgG_2a_ version (Eb_2_Mab-12-mG_2a_), the V_H_ cDNA of Eb_2_Mab-12 and the C_H_ of mouse IgG_2a_ were cloned into the pCAG-Neo vector (FUJIFILM Wako Pure Chemical Corporation (Wako), Osaka, Japan). Similarly, V_L_ cDNA of Eb_2_Mab-12 and the C_L_ of the mouse kappa chain were cloned into the pCAG-Ble vector (Wako). To construct the mouse-human IgG_1_ chimeric version (Eb_2_Mab-12-hG_1_), the V_H_ cDNA of Eb_2_Mab-12 and the C_H_ of human IgG_1_ were cloned into the pCAG-Neo vector (Wako). Similarly, V_L_ cDNA of Eb_2_Mab-12 and the C_L_ of the human kappa chain were cloned into the pCAG-Ble vector (Wako). To construct the mouse IgG_1_ version (Eb_2_Mab-12), the V_H_ cDNA of Eb_2_Mab-12 and the C_H_ of mouse IgG_1_ were cloned into the pCAG-Neo vector (Wako). Similarly, V_L_ cDNA of Eb_2_Mab-12 and the C_L_ of the mouse kappa chain were cloned into the pCAG-Ble vector (Wako). Antibody expression vectors were transfected into ExpiCHO-S cells using the ExpiCHO Expression System to produce Eb_2_Mab-12-mG_2a_, Eb_2_Mab-12-hG_1_, and Eb_2_Mab-12. As a control hIgG_1_ mAb, humCvMab-62 was generated from CvMab-62 (mouse IgG_1_, κ, an anti-SARS-CoV-2 spike protein S2 subunit mAb) using the same procedure [43]. As a control mIgG_2a_ mAb, PMab-231 (mouse IgG_2a_, κ, an anti-tiger podoplanin mAb) was previously produced [42]. All antibodies were purified using Ab-Capcher (ProteNova Co., Ltd., Kagawa, Japan).

### 4.3. Flow Cytometry

Cell lines were harvested via brief exposure to 1 mM ethylenediaminetetraacetic acid (EDTA; Nacalai Tesque, Inc.)/0.25% trypsin. After washing with 0.1% BSA in PBS (blocking buffer), the cells were treated with primary mAbs for 30 min at 4 °C, followed by treatment with anti-mouse IgG conjugated with Alexa Fluor 488 (1:2000; Cell Signaling Technology, Inc., Danvers, MA, USA) or anti-human IgG conjugated with fluorescein isothiocyanate (FITC) (1:2000; Sigma-Aldrich Corp., St. Louis, MO, USA). Fluorescence data were collected using an SA3800 Cell Analyzer (Sony Corp., Tokyo, Japan).

### 4.4. ADCC by Eb_2_Mab-12-mG_2a_ and Eb_2_Mab-12-hG_1_

Five-week-old female BALB/c nude mice were purchased from Japan SLC Inc. (Shizuoka, Japan). Spleens were aseptically excised, and single-cell suspensions were prepared by passing the tissue through a sterile cell strainer. Erythrocytes were lysed using ice-cold distilled water, and the resulting splenocytes were resuspended in culture medium, herein referred to as effector cells [56]. The ADCC activity of Eb_2_Mab-12-mG_2a_ and Eb_2_Mab-12-hG_1_ was investigated as follows. Calcein AM-labeled target cells (CHO/EphB2, MDA-MB-231, and LS174T) were co-incubated with the effector cells at an effector-to-target (E:T) ratio of 50:1 in the presence of 100 μg/mL of either control mIgG_2a_, Eb_2_Mab-12-mG_2a_, control hIgG_1_, or Eb_2_Mab-12-hG_1_. Following a 4.5 h incubation, the Calcein release into the medium was measured using a microplate reader (Power Scan HT; BioTek Instruments, Inc., Winooski, VT, USA).

Cytotoxicity was calculated as a percentage of lysis using the following formula: % lysis = (E − S)/(M − S) × 100, where E represents the fluorescence intensity from co-cultures of effector and target cells, S denotes the spontaneous fluorescence from target cells alone, and M corresponds to the maximum fluorescence obtained after complete lysis using a buffer containing 10 mM Tris-HCl (pH 7.4), 10 mM EDTA, and 0.5% Triton X-100. Data are presented as mean ± standard error of the mean (SEM). Statistical significance was evaluated using a two-tailed unpaired *t*-test.

### 4.5. CDC by Eb_2_Mab-12-mG_2a_ and Eb_2_Mab-12-hG_1_

The Calcein AM-labeled target cells (CHO/EphB2, MDA-MB-231, and LS174T) were plated and mixed with rabbit complement (final dilution 10%, Low-Tox-M Rabbit Complement; Cedarlane Laboratories, Hornby, ON, Canada) and 100 μg/mL of control mIgG_2a_, Eb_2_Mab-12-mG_2a_, control hIgG_1_, or Eb_2_Mab-12-hG_1_. Following incubation for 4.5 h at 37 °C, the Calcein release into the medium was measured, as described above.

### 4.6. Antitumor Activity of Eb_2_Mab-12-mG_2a_ and Eb_2_Mab-12-hG_1_

All animal experiments were approved and performed following regulations and guidelines to minimize animal distress and suffering in the laboratory by the Institutional Committee for Animal Experiments of the Institute of Microbial Chemistry (Numazu, Japan; approval number: 2025-021 and 2025-029). Mice were maintained on an 11 h light/13 h dark cycle in a specific pathogen-free environment across the experimental period. Food and water were supplied ad libitum. Mice weights were monitored two times per week and health were monitored three times per week. Humane endpoints for euthanasia were defined as a body weight loss exceeding 25% of the original weight and/or a maximum tumor volume greater than 3000 mm^3^.

Female BALB/c nude mice (4 weeks old) were obtained from Japan SLC, Inc. Tumor cells (0.3 mL of a 1.33 × 10^8^ cells/mL suspension in DMEM) were mixed with 0.5 mL of BD Matrigel Matrix Growth Factor Reduced (BD Biosciences). A 100 μL aliquot of the mixture, containing 5 × 10^6^ cells, was subcutaneously injected into the left flank of each mouse. To evaluate the antitumor activity of Eb_2_Mab-12-mG_2a_ and Eb_2_Mab-12-hG_1_, 100 μg of control mIgG_2a_ (*n* = 8), Eb_2_Mab-12-mG_2a_ (*n* = 8), control hIgG_1_ (*n* = 8), or Eb_2_Mab-12-hG_1_ (*n* = 8) diluted in 100 μL of PBS was administered intraperitoneally to tumor-bearing mice on day 8 post-inoculation. A second dose was administered on day 14. Mice were euthanized on day 21 following tumor cell implantation.

Tumor size was measured, and volume was calculated using the formula: volume = W^2^ × L/2, where W represents the short diameter and L the long diameter. Data are presented as the mean ± standard error of the mean (SEM). Statistical analysis was performed using one-way ANOVA followed by Sidak’s post hoc test. A *p*-value < 0.05 was considered statistically significant.

## Figures and Tables

**Figure 1 ijms-26-08302-f001:**
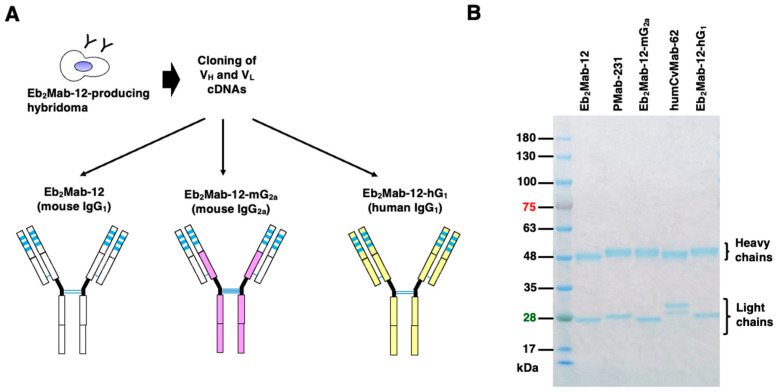
Production of recombinant Eb_2_Mab-12, Eb_2_Mab-12-mG_2a_, and Eb_2_Mab-12-hG_1_. (**A**) After determination of CDRs of Eb_2_Mab-12, recombinant Eb_2_Mab-12 (mouse IgG_1_), Eb_2_Mab-12-mG_2a_ (mouse IgG_2a_), and Eb_2_Mab-12-hG_1_ (human IgG_1_) were produced and purified. (**B**) Eb_2_Mab-12, Eb_2_Mab-12-mG_2a_, Eb_2_Mab-12-hG_1_, PMab-231 (control mIgG_2a_), and humCvMab-62 (control hIgG_1_) were treated with sodium dodecyl sulfate sample buffer containing 2-mercaptoethanol. Proteins were separated on a polyacrylamide gel. The gel was stained with Bio-Safe CBB G-250 Stain.

**Figure 2 ijms-26-08302-f002:**
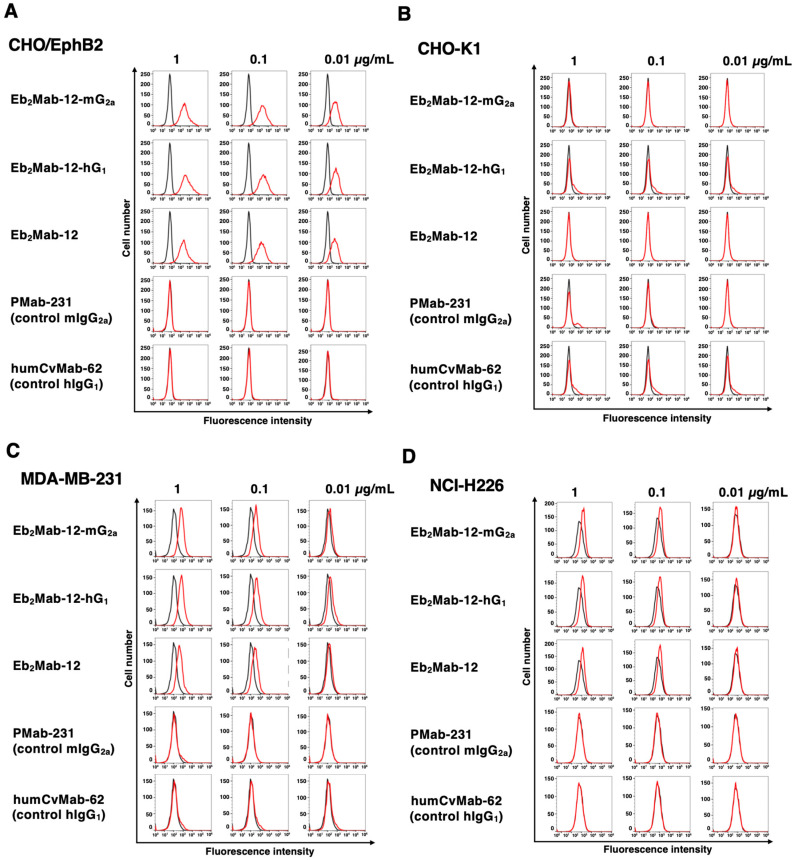
Flow cytometry analysis of Eb_2_Mab-12, Eb_2_Mab-12-mG_2a_, and Eb_2_Mab-12-hG_1_ to CHO/EphB2 and EphB2-positive cancer cell lines. CHO/EphB2 (**A**), CHO-K1 (**B**), TNBC (MDA-MB-231) (**C**), and lung mesothelioma (NCI-H226) (**D**) were treated with 0.01, 0.1, and 1 µg/mL of indicated mAbs (Red) or blocking buffer (black). Then, the cells were treated with Alexa Fluor 488-conjugated anti-mouse IgG or FITC-conjugated anti-human IgG. Fluorescence data were analyzed using the SA3800 Cell Analyzer.

**Figure 3 ijms-26-08302-f003:**
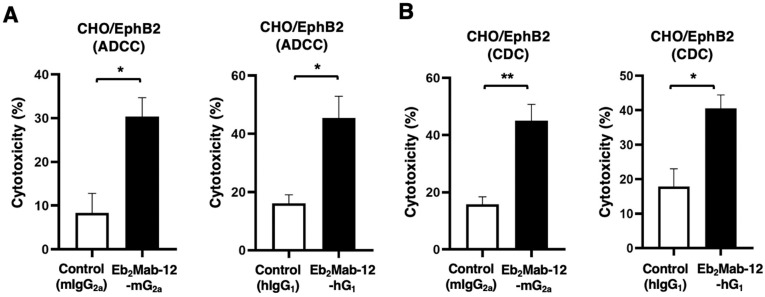
ADCC and CDC by Eb_2_Mab-12-mG_2a_ and Eb_2_Mab-12-hG_1_ against CHO/EphB2. (**A**) ADCC induced by Eb_2_Mab-12-mG_2a_ or control mouse IgG_2a_ (mIgG_2a_) against CHO/EphB2 (**left**). ADCC induced by Eb_2_Mab-12-hG_1_ or control human IgG_1_ (hIgG_1_) against CHO/EphB2 (**right**). (**B**) CDC induced by Eb_2_Mab-12-mG_2a_ or mIgG_2a_ against CHO/EphB2 (**left**). CDC induced by Eb_2_Mab-12-hG_1_ or hIgG_1_ against CHO/EphB2 (**right**). Values are shown as mean ± SEM. Asterisks indicate statistical significance (** *p* < 0.01, * *p* < 0.05; Two-tailed unpaired *t* test).

**Figure 4 ijms-26-08302-f004:**
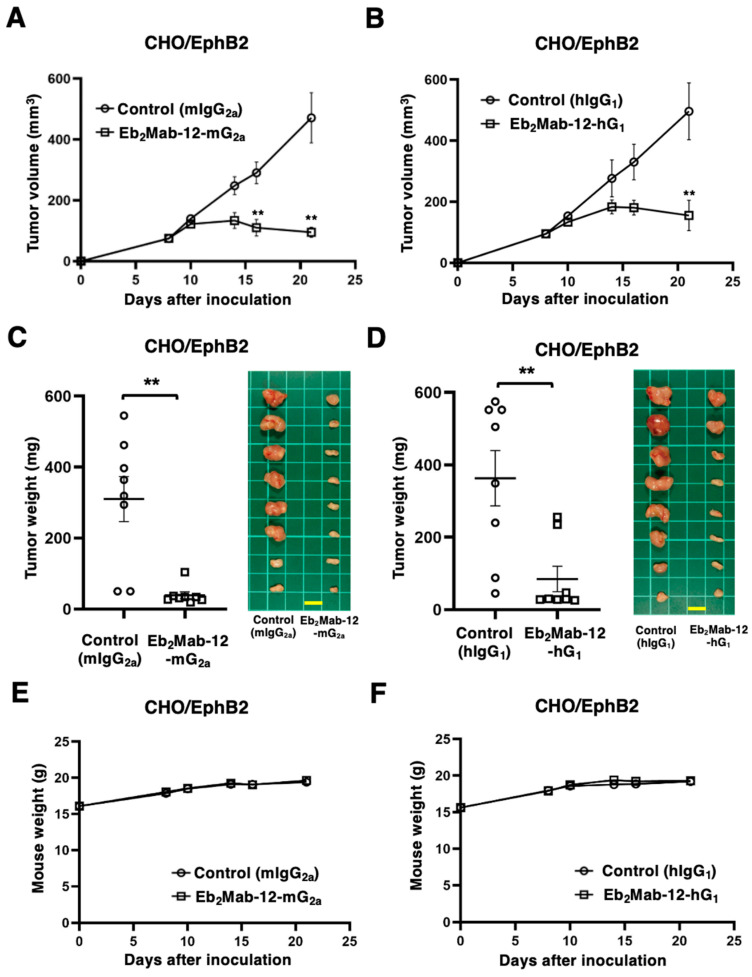
Antitumor activity of Eb_2_Mab-12-mG_2a_ and Eb_2_Mab-12-hG_1_ against CHO/EphB2 xenograft. CHO/EphB2 were subcutaneously injected into BALB/c nude mice (day 0). (**A**) In total, 100 μg of Eb_2_Mab-12-mG_2a_ or control mouse IgG_2a_ (mIgG_2a_) were intraperitoneally injected into each mouse on day 8. Additional antibodies were injected on day 14. (**B**) In total, 100 μg of Eb_2_Mab-12-hG_1_ or control human IgG_1_ (hIgG_1_) were intraperitoneally injected into each mouse on day 8. Additional antibodies were injected on day 14. The tumor volume is represented as the mean ± SEM. ** *p* < 0.01 (ANOVA with Sidak’s multiple comparisons test). (**C**,**D**) The mice treated with the mAbs mentioned above were euthanized on day 21. The CHO/EphB2 xenograft weights were measured. Values are presented as the mean ± SEM. ** *p* < 0.01 (Two-tailed unpaired *t* test). Scale bar = 1 cm. (**E**,**F**) Body weights of CHO/EphB2 xenograft-bearing mice treated with the mAbs mentioned above. There is no statistical difference.

**Figure 5 ijms-26-08302-f005:**
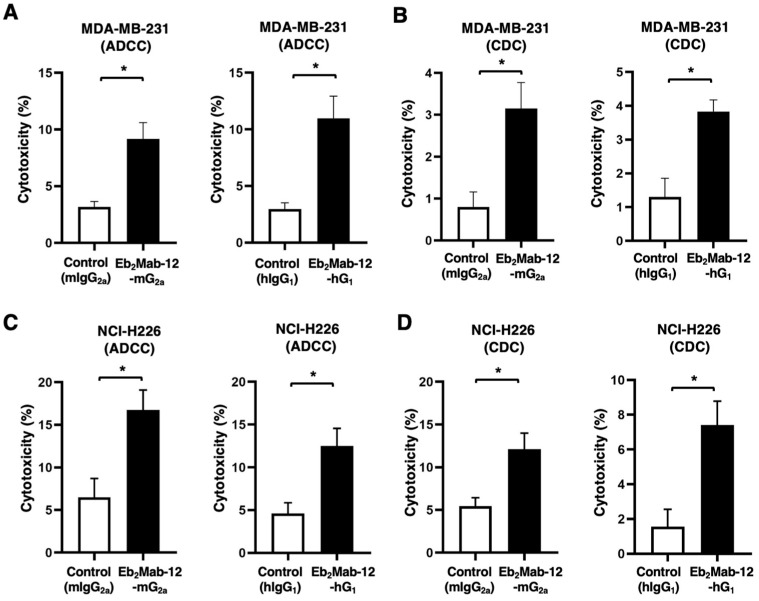
ADCC and CDC by Eb_2_Mab-12-mG_2a_ and Eb_2_Mab-12-hG_1_ against MDA-MB-231 and NCI-H226. (**A**) ADCC induced by Eb_2_Mab-12-mG_2a_ or control mouse IgG_2a_ (mIgG_2a_) against MDA-MB-231 (**left**). ADCC induced by Eb_2_Mab-12-hG_1_ or control human IgG_1_ (hIgG_1_) against MDA-MB-231 (**right**). (**B**) CDC induced by Eb_2_Mab-12-mG_2a_ or mIgG_2a_ against MDA-MB-231 (**left**). CDC induced by Eb_2_Mab-12-hG_1_ or hIgG_1_ against MDA-MB-231 (**right**). (**C**) ADCC induced by Eb_2_Mab-12-mG_2a_ or control mouse IgG_2a_ (mIgG_2a_) against NCI-H226 (**left**). ADCC induced by Eb_2_Mab-12-hG_1_ or control human IgG_1_ (hIgG_1_) against NCI-H226 (**right**). (**D**) CDC induced by Eb_2_Mab-12-mG_2a_ or mIgG_2a_ against NCI-H226 (**left**). CDC induced by Eb_2_Mab-12-hG_1_ or hIgG_1_ against NCI-H226 (**right**). Values are shown as mean ± SEM. Asterisks indicate statistical significance (* *p* < 0.05; Two-tailed unpaired *t* test).

**Figure 6 ijms-26-08302-f006:**
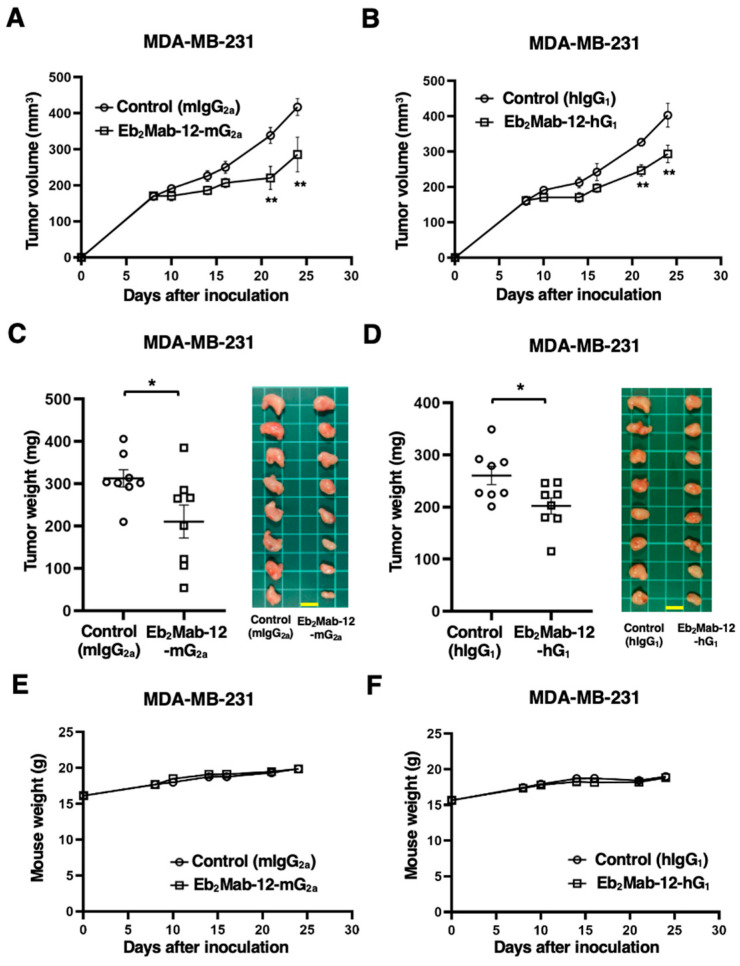
Antitumor activity of Eb_2_Mab-12-mG_2a_ and Eb_2_Mab-12-hG_1_ against MDA-MB-231 xenograft. MDA-MB-231 were subcutaneously injected into BALB/c nude mice (day 0). (**A**) In total, 100 μg of Eb_2_Mab-12-mG_2a_ or control mouse IgG_2a_ (mIgG_2a_) were intraperitoneally injected into each mouse on day 8. Additional antibodies were injected on day 14. (**B**) In total, 100 μg of Eb_2_Mab-12-hG_1_ or control human IgG_1_ (hIgG_1_) were intraperitoneally injected into each mouse on day 8. Additional antibodies were injected on day 14. The tumor volume is represented as the mean ± SEM. ** *p* < 0.01 (ANOVA with Sidak’s multiple comparisons test). (**C**,**D**) The mice treated with above-mentioned mAbs were euthanized on day 24. The MDA-MB-231 xenograft weights were measured. Values are presented as the mean ± SEM. * *p* < 0.05 (Two-tailed unpaired *t* test). Scale bar = 1 cm. (**E**,**F**) Body weights of MDA-MB-231 xenograft-bearing mice treated with above-mentioned mAbs. There is no statistical difference.

**Figure 7 ijms-26-08302-f007:**
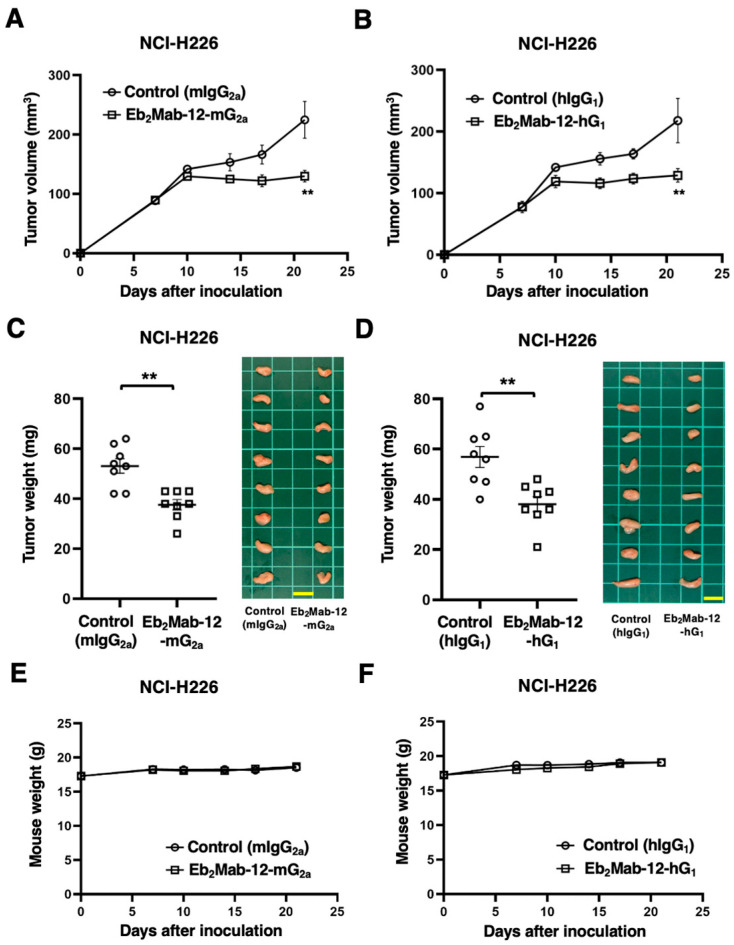
Antitumor activity of Eb_2_Mab-12-mG_2a_ and Eb_2_Mab-12-hG_1_ against NCI-H226 xenograft. NCI-H226 were subcutaneously injected into BALB/c nude mice (day 0). (**A**) In total, 100 μg of Eb_2_Mab-12-mG_2a_ or control mouse IgG_2a_ (mIgG_2a_) were intraperitoneally injected into each mouse on day 7. Additional antibodies were injected on day 14. (**B**) In total, 100 μg of Eb_2_Mab-12-hG_1_ or control human IgG_1_ (hIgG_1_) were intraperitoneally injected into each mouse on day 8. Additional antibodies were injected on day 14. The tumor volume is represented as the mean ± SEM. ** *p* < 0.01 (ANOVA with Sidak’s multiple comparisons test). (**C**,**D**) The mice treated with above-mentioned mAbs were euthanized on day 21. The NCI-H226 xenograft weights were measured. Values are presented as the mean ± SEM. ** *p* < 0.01 (Two-tailed unpaired *t* test). Scale bar = 1 cm. (**E**,**F**) Body weights of NCI-H226 xenograft-bearing mice treated with above-mentioned mAbs. There is no statistical difference.

## Data Availability

The data presented in this study are available in the article and Appendix A.

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
