# Peer review of "EphB2-Targeting Monoclonal Antibodies Exerted Antitumor Activities in Triple-Negative Breast Cancer and Lung Mesothelioma Xenograft Models"

_ijms, 2025, doi:10.3390/ijms26178302_

Round 1

Reviewer 1 Report

Comments and Suggestions for Authors

This study investigated the therapeutic potential of monoclonal antibodies targeting EphB2, a receptor overexpressed in several aggressive cancers, including triple-negative breast cancer (TNBC) and lung mesothelioma. Researchers engineered the previously developed anti-EphB2 antibody Eb2Mab-12 into mouse IgG2a (Eb2Mab-12-mG2a) and human IgG1 (Eb2Mab-12-hG1) formats, both retaining high specificity for EphB2-positive cells. These antibodies exhibited strong antibody-dependent cellular cytotoxicity (ADCC) and complement-dependent cytotoxicity (CDC) against EphB2-expressing cell lines. In xenograft models of TNBC, lung mesothelioma, and CHO/EphB2 cells, both antibodies significantly inhibited tumor growth without causing notable body weight loss. However, limited efficacy was observed against rapidly growing LS174T colorectal cancer xenografts despite strong in vitro binding. The findings suggest that Eb2Mab-12-derived antibodies could be promising candidates for mAb-based therapy in EphB2-positive tumors, warranting further preclinical and clinical evaluation.

Overall, I find this to be a well-conducted and interesting study that provides valuable insights. The experimental results are compelling, and the manuscript is clearly written. After addressing the following points, I would support its publication.

  1. Perform immune profiling of treated tumors (e.g., flow cytometry or immunohistochemistry for NK cells, macrophages, and T cells) to directly link antitumor effects to immune mechanisms.
  2. Add a dose–response study for Eb2Mab-12-mG2a and Eb2Mab-12-hG1 to determine minimal effective doses and potential toxicity thresholds.
  3. Although body weight stayed stable, that’s a pretty rough metric for safety. Without PK, biodistribution, or toxicity data, it’s hard to assess the real translational potential, especially for humanized formats.
  4. There’s little discussion of how these antibodies compare to small-molecule inhibitors or antibody-drug conjugates in terms of efficacy, specificity, or potential resistance mechanisms.

Reviewer 2 Report

Comments and Suggestions for Authors

The Introduction discusses ependymomas, but not mesotheliomas. Please, provide a brief introduction regarding mesotheliomas.

The Discussion does not discuss the data with the NCI-H266 mesothelioma cell line. Please correct this omission.

Also, a brief paragraph is needed for explicitly stating the caveats of xenografts in nude mice regarding their value as preclinical data, especially that humanized Onco-Hu BALB/c mouse models are available to use with human cancer cell lines.

In the Materials and Methods section on page 14, line 395, reference 43 cited as a source for preparing the splenocytes doesn't contain this info, but refers to yet another source, which also lacks the actual method for preparing the splenocytes. Please, provide a valid reference + a brief description of the actual method used for preparing the splenocytes.
